# Sugar Levels Determine Fermentation Dynamics during Yeast Pastry Making and Its Impact on Dough and Product Characteristics

**DOI:** 10.3390/foods11101388

**Published:** 2022-05-11

**Authors:** Evelyne Timmermans, An Bautil, Kristof Brijs, Ilse Scheirlinck, Roel Van der Meulen, Christophe M. Courtin

**Affiliations:** 1Laboratory of Food Chemistry and Biochemistry, Leuven Food Science and Nutrition Research Centre (LFoRCe), KU Leuven, Kasteelpark Arenberg 20, 3001 Leuven, Belgium; evelyne.timmermans@kuleuven.be (E.T.); an.bautil@kuleuven.be (A.B.); kristof.brijs@kuleuven.be (K.B.); 2Vandemoortele Izegem NV, Prins Albertlaan 12, 8870 Izegem, Belgium; ilse.scheirlinck@vandemoortele.com (I.S.); roel.vandermeulen@vandemoortele.com (R.V.d.M.)

**Keywords:** yeast, sugar, fermentation, pastry, fermentation metabolites, dough rheology, osmotic stress

## Abstract

Fermented pastry products are produced by fermenting and baking multi-layered dough. Increasing our knowledge of the impact of the fermentation process during pastry making could offer opportunities for improving the production process or end-product quality, whereas increasing our knowledge on the sugar release and consumption dynamics by yeast could help to design sugar reduction strategies. Therefore, this study investigates the impact of yeast fermentation and different sugar concentrations on pastry dough properties and product quality characteristics. First, yeasted pastry samples were made with 8% yeast and 14% sucrose on a wheat flour dry matter base and compared to non-yeasted samples. Analysis of saccharide concentrations revealed that sucrose was almost entirely degraded by invertase in yeasted samples after mixing. Fructans were also degraded extensively, but more slowly. At least 23.6 ± 2.6% of the released glucose was consumed during fermentation. CO_2_ production during fermentation contributed more to product height development than water and ethanol evaporation during baking. Yeast metabolites weakened the gluten network, causing a reduction in dough strength and extensibility. However, fermentation time had a more significant impact on dough rheology parameters than the presence of yeast. In balance, yeast fermentation did not significantly affect the calculated sweetness factor of the pastry product with 14% added sucrose. Increasing the sugar content (21%) led to higher osmotic stress, resulting in reduced sugar consumption, reduced CO_2_ and ethanol production and a lower product volume. A darker colour and a higher sweetness factor were obtained. Reducing the sugar content (7%) had the opposite effect. Eliminating sucrose from the recipe (0%) resulted in a shortened productive fermentation time due to sugar depletion. Dough rheology was affected to a limited extent by changes in sucrose addition, although no sucrose addition or a very high sucrose level (21%) reduced the maximum dough strength. Based on the insights obtained in this study, yeast-based strategies can be developed to improve the production and quality of fermented pastry.

## 1. Introduction

Yeast is used worldwide as a leavening agent in fermented baked goods such as bread and Danish pastry. Most often, *Saccharomyces cerevisiae*, also known as baker’s yeast, is used [1]. During alcoholic fermentation, yeast converts sugars into carbon dioxide (CO_2_) and ethanol [1], with a preference for glucose consumption over fructose and maltose. This yeast-mediated fermentation also yields secondary metabolites such as glycerol, organic acids and aroma compounds. These fermentation metabolites play an important role in dough properties and product quality characteristics [2,3,4]. Over the past decade, several studies have focused on the role of yeast during fermentation in breadmaking [2,3,4,5,6,7,8,9,10]. It is well established that CO_2_ and ethanol cause an increase in bread loaf volume. Next to CO_2_, organic acids and glycerol released by yeast during fermentation influence dough rheology and the shelf life of baked goods [3,11,12,13]. In recent years, there has been more awareness that yeast is not only responsible for dough leavening, but also for the production of many aromatic secondary metabolites, such as esters, aldehydes and ketones, produced during alcoholic fermentation, that greatly contribute to the flavour profile of bakery products [14]. Not only the metabolites and aroma compounds produced upon fermentation by yeast, but also the yeast-mediated sugar release and consumption, can influence dough properties and product quality characteristics. Although the exact mechanism is still unclear, sucrose and other low molecular weight sugars, like glucose, in aqueous media increase both starch gelatinisation and the protein denaturation temperature [15,16,17]. This increase in gelatinisation temperature can cause a change in the volume and texture of baked products [18]. Moreover, high sucrose levels are known to inhibit gluten hydration and cross-linking [17,19,20].

The release and consumption of different sugars by yeast also influence the sweetness and colour of baked products. The most abundant mono- and disaccharides in white wheat flour are sucrose (2.16 ± 0.26 mg/g of flour), fructose (0.91 ± 0.13 mg/g of flour), maltose (0.53 ± 0.09 mg/g of flour) and glucose (0.54 ± 0.09 mg/g of flour) [6]. In the presence of yeast, sucrose itself does not contribute to sweetness since it is rapidly hydrolysed by yeast invertase into glucose and fructose [9]. Glucose and fructose have an average relative sweetness of 0.7 and 1.5, respectively, compared to sucrose, which is the reference for calculating the sweetness of a product [21,22,23,24]. Besides glucose and fructose, other sugars present in the flour, such as maltose and raffinose, and fermentation products, such as polyols, can also have a pronounced influence on the sweetness of a product. Maltose and raffinose have an average relative sweetness of 0.50 and 0.22, respectively. The relative sweetness of polyols varies between 0.58 (mannitol) and 1.12 (xylitol) [21,22,23,24,25]. Furthermore, reducing sugars, like glucose and fructose, also contribute to colour formation during baking due to Maillard and caramelisation reactions [20]. We can safely state that fermentation has a big impact on the end-product quality of fermented baked goods, given the numerous effects of yeast-mediated changes in sugar levels and the production of fermentation metabolites on product quality.

Despite this considerable impact, little research has focused on the role of yeast during the fermentation of pastry products, like Danish pastry and ‘croissants’. The role of fermentation in layered pastry products is partially different than that in other bakery products. The produced CO_2_ is, together with the layered structure, responsible for the characteristic texture of fermented pastry. This flaky texture is achieved by producing and baking multi-layered dough, which consists of alternating layers of dough and bakery fat. The latter can be margarine, butter or shortening [26,27,28]. Essential ingredients for fermented pastry are water, refined (wheat) flour, bakery fat, yeast and salt. Typically, sucrose, milk powder, eggs and improvers are also part of the recipe [29,30]. Dough is prepared by mixing the ingredients into a ‘predough’, in which a sheet of bakery fat is folded [29]. This dough with one layer of fat is then repeatedly sheeted and folded, resulting in a multi-layered dough-fat system. Research on this kind of system has mainly focused on the functionality of fat [31,32,33,34,35], wheat flour gluten proteins [36,37] and wheat flour starch [37]. However, the role of yeast, the fate of sugar and the impact of rapid sucrose hydrolysis on yeast during fermentation and baking has not been studied in detail for these multi-layered dough-fat systems. 

This study aims to investigate fermentation dynamics during pastry making and the impact of sugar, yeast and the fermentation process on pastry dough properties and product quality characteristics. This is achieved by analysing the sugar release and production, metabolite production and end-product quality in terms of dough texture, product height and sweetness factor in unyeasted pastry samples and yeasted pastry samples with different sugar concentrations. Increasing our knowledge on the impact of the fermentation process during pastry making could offer opportunities to improve the production process or end-product quality, whereas increasing our knowledge on the sugar release and consumption dynamics by yeast could help to design sugar reduction strategies. By adapting the fermentation phase, it can, for example, be possible to reduce the sugar content in fermented pastry. Reduced sugar content is of interest since excess sugar consumption is considered a health risk and because there is an increasing trend in consumer awareness towards healthy foods.

## 2. Materials and Methods

### 2.1. Materials

Commercial wheat flour (protein level: 13.8 ± 0.2%, dry matter (dm) basis; moisture content: 13.7 ± 0.1%) was obtained from Paniflower (Merksem, Belgium). The protein level (N × 5.7) was determined in duplicate according to an adaptation of the AOAC method 990.03 to an automated Dumas protein analysis system (VarioMax Cube N, Elementar, Hanau, Germany). The moisture content was analysed using AACC International Approved Method 44-15.02.23. Palm- and rapeseed-based margarine (80% fat, free from emulsifiers) was provided by Vandemoortele (Izegem, Belgium). Commercial traditional baker’s yeast (*Saccharomyces cerevisiae*) in compressed form was obtained from Algist Bruggeman (Gent, Belgium), fine sucrose from Tiense Suikerraffinaderij (Tienen, Belgium), sodium chloride from Everyday (Halle, Belgium) and ascorbic acid from Aland (Jiangsu, China) Nutraceutical Co., Ltd. All reagents, solvents and chemicals were of analytical grade and obtained from Sigma-Aldrich (Bornem, Belgium) unless indicated otherwise.

### 2.2. Pastry Sample Preparation

Fermented pastry samples were produced using a laboratory-scale procedure according to Ooms et al. [36]. First, a predough was prepared by mixing wheat flour (500.0 g), margarine (25.0 g), tap water (250.0 g), sucrose (60.0 g), compressed fresh yeast (35.0 g), sodium chloride (10.0 g) and ascorbic acid (150.0 mg) at room temperature (23 °C) for 7 min using a Vema Construct mixer (Izegem, Belgium). Then, the dough thickness was reduced to 7 mm in six steps using a Rondo STM 513 laminating device (Burgdorf, Switzerland), after which roll-in/laminating margarine (25.0% on dough weight) was folded into the predough according to the envelope method [27,29]. A lower temperature for the margarine (16 °C) than for the predough (23 °C) was maintained to ensure that the roll-in margarine did not melt during the lamination step. Next, the dough was sheeted in seven steps until a final thickness of 5 mm was obtained. After the third step, the dough was turned over 90 degrees to limit excessive shrinkage in one direction during baking. After the last sheeting step, the dough was folded into three, yielding three fat layers (Figure 1a). These sheeting and folding steps were repeated until a multi-layered dough with nine fat layers was obtained. Next, the multi-layered dough was allowed to rest at 4 °C for 20 min. This resting avoids softening/melting of the margarine and allows for the relaxation of the gluten network, which is necessary to limit shrinkage during baking [29]. After resting, the dough was again sheeted and folded until 27 fat layers were obtained, whereafter the dough was again sheeted to its final thickness (5 mm) in seven steps. Finally, the dough was cut into pieces using a squared cutter with a squared hole in the middle. This geometry was chosen to obtain reproducible products for analysis afterwards (Figure 1b). Fermentation of the squared dough pieces was performed in a fermentation cabinet (30 °C, 80% relative humidity, Hein Condilux, Strassen, Luxemburg) for 75 min. Finally, the samples were baked in a conduction oven (Hein Condilux) with a top temperature of 220 °C and a bottom temperature of 210 °C for 16 min. To investigate yeast and sugar functionality, unyeasted samples with 60 g sucrose (14% *w*/*w* dm flour) and yeasted samples with 0 g, 30 g (7% *w*/*w* dm flour) and 90 g (21% *w*/*w* dm flour) sucrose content were prepared in the same way as the reference samples (60 g (14% *w*/*w* dm flour) sucrose and 35 g (8% *w*/*w* dm flour) yeast). For the analysis of sugars and metabolites, no margarine was added to the recipe. This did not affect the fermentation and allowed us to safeguard the chromatography equipment. To compare the colour between the different samples, pictures of the samples were taken after baking (Figure 1b).

### 2.3. Quantification of Mono-, Di- and Trisaccharides and Fructan Content

Dough and pastry samples were taken after mixing, during lamination, fermentation and during and after baking. The samples were immediately frozen in liquid nitrogen, lyophilised and ground with a mortar and pestle to obtain a powder (moisture content: 3–4%). To quantify the saccharide content, samples were first defatted with hexane to safeguard the equipment used for sugar and metabolite analysis. This step was omitted when samples were made without in-dough and roll-in margarine (cf. Section 2.2). For defatting, samples (1.0 g) were shaken (30 min, 150 rpm, room temperature) in 10.0 mL hexane. After shaking, the hexane phase was removed. This extraction with hexane was repeated twice. The resulting pellet was dried under a fume hood overnight. Next, an enzyme deactivation step was also performed on the pellet to avoid further degradation of sugars. A total of 1.5 mL of an internal standard solution (8.0 mg rhamnose/mL) was added to 500 mg of (defatted) sample. Sugars were extracted as described by Struyf et al. [9]. The extracts were diluted to obtain a final internal standard concentration of 10 µg/mL and stored at −20 °C until further analysis. Saccharides were separated by High-Performance Anion-Exchange Chromatography with integrated Pulsed Amperometric Detection (HPAEC-iPAD) on a Dionex ICS5000 chromatography system (Thermo Fisher Scientific, Sunnyvale, CA, USA). A CarboPac PA-100 guard column (4 × 50 mm) (Thermo Fisher Scientific, Sunnyvale, CA, USA) coupled to a CarboPac PA-100 column (4 × 250 mm) (Thermo Fisher Scientific, Sunnyvale, CA, USA), equilibrated with 90 mM NaOH, with a sodium acetate gradient (0–6 min, 10 mM; 6–16 min, 10–100 mM; 16–20 min, 100–175 mM) and a flow of 1 mL/min, was used. Rhamnose was used as an internal standard to identify and quantify the saccharides. 

Fructan concentrations in wheat flour and the (defatted) lyophilised dough powders were determined after the samples were subjected to an enzyme deactivation step and mild acid hydrolysis as described by Verspreet et al. [38]. Measurements of saccharides in wheat flour and pastry dough samples were performed in triplicate. Measurements of fructan concentrations were done in duplicate.

Saccharide and fructan concentrations are expressed as weight percentages on flour dry matter base (% *w*/*w* dm).

### 2.4. Quantification of Ethanol, Glycerol and Organic Acids Content

To quantify the metabolites initially present in flour or produced by yeast during pastry making, flour, dough and pastry samples (15.0 g) were taken before and after mixing, during sheeting and folding, during fermentation and after baking. The metabolites were extracted by blending the sample immediately with deionised water (two times the amount of the dough sample) with a Waring 8011E blender (Waring Products, Torrington, CT, USA) for 30 s [39]. A total of 1.5 mL of each blended sample was subsequently centrifuged (11,000 rpm, 3 min, 23 °C) with an Eppendorf centrifuge 5415D (Eppendorf AG, Hamburg, Germany). The supernatant was filtered with a Millex-HP 0.22 µm polyethersulfone membrane and stored at −20 °C until further analysis. Ethanol, glycerol, acetic acid and succinic acid in the dough extracts were analysed with Ion-Exclusion High-Performance Liquid Chromatography (IE-HPLC) using an LC-20AT modular HPLC system (Shimadzu, Kyoto, Japan). The metabolites were separated at 60 °C using an ion-exclusion ROA-organic acids guard (50 × 7.8 mm) and analytical column (300 × 7.8 mm) (Phenomenex, Torrance, CA, USA) with an H2SO4 solution (2.50 mM) as mobile phase and a flow rate of 0.60 mL/min. The detection was with a Refractive Index Detector 10A (Shimadzu, Kyoto, Japan). Measurements were performed in duplicate, starting from two biological replicates.

### 2.5. CO_2_ Production Analysis

Dough samples were prepared with the same ingredients and concentrations described in Section 2.2 but on a 10 g flour scale. The dough was not laminated because part of the produced CO_2_ would escape during laminating. No margarine was added. The ingredients were mixed for 4 min with a 10 g pin bowl mixer (National Manufacturing, Lincoln, NE USA). Immediately after mixing, the dough samples were fermented in Risograph canisters (400 mL, National Manufacturing, Lincoln, NE, USA) at 30 °C for 180 min. The CO_2_ production in each canister was measured every minute by the Risograph (National Manufacturing, Lincoln, NE, USA). Measurements were performed in triplicate, starting from three biological replicates.

### 2.6. Pastry Burst Rig Texture Measurements

Dough strength, elasticity and extensibility were analysed on laminated dough samples with margarine, prepared as described in Section 2.2, using a pastry burst rig (Stable Micro Systems, Godalming, Surrey, UK) with an Instron 3342 texture analyser (Norwood, MA, USA) equipped with a 5.0 kg load cell. Before and during fermentation, dough pieces (10 × 10 cm) were cut with a knife and fixed between two metal plates to obtain circular (6.5 cm diameter) sample sections. During analysis, a spherical probe (2.5 cm diameter) moved down at a constant speed of 2.0 mm/s. The load (N) was measured in function of dough extension (mm). From these results, dough elasticity (slope of the linear part, N/mm), strength (maximum load, N) and extensibility (maximum extension, mm) were calculated. Measurements were performed in triplicate, starting from three biological replicates.

### 2.7. Dough and End-Product Height Measurement

Before, during and after fermentation and after baking, the dough sample and end-product height were measured using a ruler. Measurements were performed in triplicate starting from three biological replicates.

### 2.8. Calculation of The Sweetness of the End-Product

The sweetness factor of pastry was calculated according to Moskowitz [39], who proposed a formula to calculate the sweetness of a sugar solution:S = kC^n^,
where S is the sweetness, k is the relative sweetness of the sugar compared to sucrose, C is the concentration of the sugar and n is an exponent equated to 1.3. Using this formula, the sweetness of each sugar in a fermented and baked pastry sample was calculated. The sweetness factor of yeasted and unyeasted pastry samples was equated to the sum of the sweetness of the individual components in the product. The relative sweetness values of the components that were used are averages of the relative sweetness values found in the literature (sucrose: 1.00; glucose: 0.70; fructose: 1.50; raffinose: 0.22; maltose: 0.50) [21,22,23,24]. From related experiments with a trained sensory panel, we know there is a strong correlation between the calculated sweetness factor and the sweetness perceived by a trained sensory panel.

### 2.9. Statistical Analysis

Data were analysed using the Fit Model platform of the statistical software JMP Pro 15 (SAS Inst., Cary, NC, USA). Mean values of yeasted and unyeasted samples were compared using the Student’s *t*-test, while mean values of yeasted samples with different sugar concentrations were compared using a one-way analysis of variance (ANOVA), followed by the post hoc Tukey HSD test. Mean differences were considered significant when *p* < 0.05.

## 3. Results

### 3.1. Impact of Yeast and Fermentation Dynamics in Pastry Dough

#### 3.1.1. Sugar Release during Fermented Pastry Production

The effect of yeast on the evolution of saccharide concentrations during the production of pastry samples was investigated by analysing yeasted and unyeasted samples, both with 14% added sucrose, with HPAEC-iPAD. To protect the chromatography devices, pastry samples have to be defatted before analysis. If sugar release and consumption are not affected by the high amount of margarine in fermented pastry, samples for chromatography can be prepared without margarine, and the defatting step can be omitted. Therefore, first, the evolution of glucose, fructose, maltose, melibiose and raffinose concentrations in samples without roll-in and in-dough margarine and samples with only in-dough margarine were compared with the evolution in samples with in-dough and roll-in margarine. No significant differences were observed in saccharide concentrations between both types of samples (Appendix A), implying that margarine does not impact the yeast-mediated sugar release and consumption during fermented pastry making. Given this observation, we also assumed that the production of metabolites would not be affected by the presence of margarine. As a result, samples for sugar and metabolite analysis were prepared without in-dough and roll-in margarine.

The evolution of sucrose, glucose, fructose and maltose concentrations during pastry making in unyeasted and yeasted dough samples is shown in Figure 2. In the unyeasted samples, the sucrose concentration decreased slightly from 14.13 ± 0.01% to 12.88 ± 0.23% and glucose, fructose and maltose concentrations increased from 0.02 ± 0.01% to 0.19 ± 0.01%, from 0.03 ± 0.00% to 0.12 ± 0.00% and from 0.04 ± 0.01% to 1.70 ± 0.10%, respectively. The addition of yeast to the pastry dough resulted, mainly during mixing, in a very rapid decrease in the sucrose concentration from 14.13 ± 0.01% to 0.00 ± 0.00%, because of hydrolysis by yeast invertase. This contributed largely to an increase in glucose and fructose concentrations, from 0.02 ± 0.01% to 7.33 ± 0.25% and from 0.03 ± 0.00% to 7.97 ± 0.26%, respectively. The maltose concentration increased from 0.04 ± 0.01% to 1.32 ± 0.08%, due to amylase activity. At every step in the pastry production process, sugar concentrations between yeasted and unyeasted samples were significantly different (*p* < 0.001).

#### 3.1.2. Degradation of Fructan during Fermented Pastry Production

The evolution of the fructan concentration during pastry production was analysed in yeasted and unyeasted dough samples without added sucrose (Figure 3). In unyeasted pastry samples, the fructan concentration decreased slightly from 1.25 ± 0.02% to 1.08 ± 0.01%, while in yeasted pastry samples, the fructan concentration was lowered considerably during the first hour of the process, from 1.25 ± 0.02% to 0.18 ± 0.03%. Due to the interference of high sugar concentrations with the fructan analysis, the fructan content could not be analysed in samples with added sucrose. 

#### 3.1.3. Production of Fermentation Metabolites during Fermented Pastry Making

The CO_2_ production rate (mL CO_2_/min in 10 g flour dough) was measured in pastry dough prepared without margarine, as margarine did not affect the fermentation capacity (Section 3.1) (Figure 4). As expected, in unyeasted dough, no CO_2_ was produced. In yeasted dough, a total of 157.7 ± 0.4 mL CO_2_ was produced after 3 h of fermentation at 30 °C.

Figure 5 shows the evolution of ethanol, glycerol and acetic and succinic acid concentrations during pastry making in unyeasted and yeasted pastry samples. Low concentrations of succinic acid and glycerol were present in unyeasted samples at the end of the production process (0.04 ± 0.002% (*w*/*w* dm flour) and 0.05 ± 0.003%, respectively). Acetic acid and ethanol were not present. At every step in the pastry production process, metabolite concentrations in yeasted samples were significantly higher (*p* < 0.001) than in unyeasted samples. In yeasted samples, 1.23 ± 0.03% ethanol was measured after fermentation. During baking, almost all ethanol was evaporated, resulting in a concentration of 0.11 ± 0.03% ethanol in the end-product. A total of 0.72 ± 0.03% glycerol was produced by yeast, mainly during fermentation and baking. The production of succinic acid (0.16 ± 0.02%) and acetic acid (0.05 ± 0.01%) caused a decrease in pH from 5.75 ± 0.04 to 5.23 ± 0.10 during the production process. 

### 3.2. The Impact of Fermentation on Pastry Dough and End-Product Properties

#### 3.2.1. Pastry Dough and End-Product Height

Figure 6 shows the increase in the height of unyeasted and yeasted pastry samples during the production process. In unyeasted samples, the height only increased during baking, while in yeasted samples, the height increased during fermentation and baking. From a fermentation period of 30 min onwards, the height between yeasted and unyeasted samples was significantly different (*p* < 0.001).

#### 3.2.2. Pastry Dough Rheology

In Figure 7, changes in dough strength, extensibility and elasticity of yeasted and unyeasted laminated pastry dough samples during an incubation period of 75 min at 30 °C and 80% relative humidity are shown. As a function of time, the dough strength and elasticity decreased in both samples, while the dough extensibility increased. The maximum dough strength was consistently lower in yeasted samples than in unyeasted samples. In yeasted samples, the increase in extensibility was smaller than in unyeasted samples (Figure 7). There was no significant difference in elasticity between yeasted and unyeasted samples (*p* > 0.05).

#### 3.2.3. Pastry Sweetness

The sweetness factor of unyeasted pastry samples and yeasted pastry samples with different sucrose concentrations is shown in Table 1. The sweetness factor of unyeasted samples was not significantly different from the sweetness factor of yeasted samples with 14% added sucrose (*p* = 0.428). However, in unyeasted pastry samples, sucrose had by far the most significant contribution (96.0%) to the sweetness factor, whereas, in yeasted pastry samples, fructose had the most significant contribution (73.2%).

### 3.3. Impact of Sucrose Level on Yeast Activity in Fermented Pastry Making

#### 3.3.1. Sugar Release

The effect of different sucrose addition levels on the evolution of saccharide concentrations during the production of pastry samples was investigated by analysing samples with an increasing sucrose addition amount with HPAEC-iPAD (Figure 2). In samples without added sucrose, the highest glucose and fructose concentrations during the production process were equal to 0.36 ± 0.06% and 0.85 ± 0.12% (on dm flour base), respectively. When sucrose addition was increased with steps of 7%, a corresponding increase in glucose and fructose release was visible. In samples with 21% added sucrose, the maximum glucose concentration was equal to 10.33 ± 1.24%, and the maximum fructose concentration was equal to 10.82 ± 1.02%. Increasing the sucrose addition led to a reduced maltose concentration during the first 60 min of the production process and an increased maltose concentration in the end-product.

#### 3.3.2. Sugar Consumption

From the results in Figure 2, sugar consumption values on a dm flour base were calculated. In the samples with the highest sucrose addition (21%), the lowest amount of sugar was consumed (0.90% glucose). However, these consumption amounts are minima, as the dynamics of mono- and disaccharides being formed by enzyme activity and at the same time being consumed by yeast cannot be captured in these single concentration values. Reducing the sucrose addition resulted in a higher sugar consumption: at least 1.79% (*w*/*w* dm flour) glucose, 0.49% fructose and 0.14% maltose in samples with 14% added sucrose; and 2.64% glucose, 1.21% fructose and 0.30% maltose in samples with 7% added sucrose. When the sucrose addition was further reduced to 0%, the total sugar consumption was not significantly different from the samples with 14% added sucrose, but maltose became the most consumed sugar instead of glucose. In these samples, at least 0.33% glucose, 0.84% fructose and 1.37% maltose was consumed.

#### 3.3.3. Fermentation Metabolite Production

The CO_2_ production rate (mL CO_2_/min in 10 g flour dough) was measured in fermented pastry dough with increasing sucrose addition amounts (0, 7, 14 and 21% on dm flour base) (Figure 4). In dough without added sucrose, the CO_2_ production rate was high at the beginning of fermentation but dropped significantly after ±90 min. After 3 h, a total of 107.9 ± 1.0 mL CO_2_ was produced in these samples. When the sucrose addition was stepwise increased by 7%, there was no drop in total CO_2_ production during the whole measurement time, but the CO_2_ production rate and total CO_2_ production volume decreased. In samples with 7, 14 or 21% added sucrose, a total of 203.7 ± 2.9 mL, 157.7 ± 0.4 mL or 94.3 ± 4.8 mL CO_2_ was produced, respectively, after 3 h of fermentation at 30 °C.

Figure 5 shows the evolution of ethanol, glycerol and acetic and succinic acid concentrations during pastry making in samples with and without added sucrose. When no sucrose was added, an increased ethanol content (1.37 ± 0.12%) was measured after fermentation, whereas the glycerol content in the end-product (0.51 ± 0.06) was significantly lower compared to the samples with 14% added sucrose. The production of succinic acid (0.15 ± 0.00%) and acetic acid (0.04 ± 0.01) in the samples without added sucrose was not significantly different from the organic acid production in the samples with 14% sucrose.

### 3.4. The Impact of Increasing Sucrose Levels on Dough and End-Product Characteristics in Fermented Pastry Making

#### 3.4.1. Pastry Dough and End-Product Height

Figure 6 shows the impact of sucrose on both dough and end-product height. When sucrose addition levels were increased, the sample height increased less during fermentation. However, after fermentation and after baking, there was no significant difference in the height of samples with 0, 7 or 14% added sucrose. When 21% sucrose was added, both after fermentation (*p* = 0.013) and baking (*p* = 0.004), the end-product height was significantly lower than samples with lower sucrose concentrations.

#### 3.4.2. Pastry Dough Rheology

In Figure 7, changes in dough strength, extensibility and elasticity of laminated pastry dough samples with an increasing sucrose addition (0, 7, 14 or 21% on dm flour base) during an incubation period of 75 min at 30 °C and 80% relative humidity are shown. There was no significant difference in dough rheology between samples with 7 or 14% added sucrose. The lowest (0%) and highest (21%) sucrose levels led, however, to a significantly lower maximum dough strength, especially during the first 25 min of fermentation.

#### 3.4.3. Pastry Sweetness

When the sucrose addition was reduced, a lower sweetness factor was obtained, while an increased sucrose addition resulted in a higher sweetness factor.

## 4. Discussion

The fermentation characteristics and role of yeast and sugar during the fermentation phase of breadmaking has been widely studied over the past few decades. However, to our knowledge, a clear view on fermentation and the interplay between yeast and sugar during fermented pastry making is lacking. Therefore, in this study, we investigated the dynamics of fermentation, the impact of increasing sucrose levels on these fermentation dynamics, and the impact of yeast fermentation and sucrose levels on dough properties and end-product quality characteristics in fermented pastry making.

### 4.1. Fermentation Dynamics in Pastry Dough

To evaluate the fermentation dynamics in pastry dough, samples with and without yeast were made and the levels of sugars and metabolites followed up during the fermentation phase. 

In yeasted pastry samples, significant changes in sugar concentrations were observed. Due to the action of the yeast enzyme invertase [1], all sucrose, added and endogenously present, as well as part of the fructan present in the flour, were hydrolysed into glucose and fructose, mainly during the mixing step. These results are in line with those from Nilsson et al. [40], Verspreet et al. [38] and Struyf et al. [10], who observed a fast degradation of sucrose and fructan in bread dough samples. Glucose is the preferred substrate of yeast [1,41]. In this study, at least 23.6 ± 2.6% of the total amount of glucose released from the sucrose or fructan was consumed. The residual glucose and fructose remained in the final product. Since the maltose metabolism in *S. cerevisiae* is regulated by glucose repression, the consumption of maltose is much lower compared to glucose (and fructose) [42]. However, in yeasted samples, maltose concentrations were lower than in unyeasted samples. Possibly a small amount of maltose was consumed by yeast, although an abundant amount of glucose and fructose was still present in the dough after 75 min of fermentation.

In unyeasted pastry samples, only minor changes in sugar concentrations were observed. The slight decrease in sucrose and increase in glucose and fructose concentrations in unyeasted samples may be attributed to sucrose hydrolysis by microbial enzymes present in flour, such as invertases, glucoamylases or α-glucosidases [43]. Possibly, part of the released glucose and fructose participated in Maillard and caramelisation reactions during baking or was consumed by contaminating microorganisms. Cardoso et al. [44] found several microorganisms in wheat flours that can be responsible for low levels of sugar consumption. The increase in the maltose concentration results from starch hydrolysis by endogenous flour α-amylases [1]. These results are consistent with results obtained by Potus et al. [45], Struyf et al. [43] and Struyf et al. [10] in unyeasted bread dough.

Consumed sugars are converted into CO_2_, ethanol and other metabolites. In yeasted samples, high CO_2_ production rates were observed. However, these CO_2_ production rates were still lower than the CO_2_ production rates in bread dough during a fermentation of 120 min [9]. This is probably due to the higher sugar concentration in pastry samples, which leads to more osmotic stress in yeast cells. Indeed, several studies have already reported that osmotic stress results in a reduced fermentation rate and, consequently, reduced CO_2_ production [46]. Furthermore, compared to the CO_2_ production in bread dough [9], which rapidly decreased after 2 h of fermentation, the CO_2_ production in fermented pastry increased for a longer time (at least 3 h) due to the higher sucrose amount added to pastry dough compared to bread dough. 

Next to CO_2_, ethanol and other secondary metabolites were produced in yeasted samples. Yeast produced, however, less ethanol in pastry dough samples with 8% yeast (1.23 ± 0.03%) than in bread dough with 5.3% yeast [5]. Similar to the lower CO_2_ production rate, this observation can be explained by the reduced yeast activity as a result of the higher osmotic stress in pastry dough samples [46]. To protect themselves against this osmotic stress, yeast cells produce and accumulate glycerol [47]. In pastry samples with 8% yeast, the glycerol concentration (0.80 ± 0.06%) was not higher than in bread samples made with 5.3% yeast [5]. Probably, the glycerol production in bread was also already increased to protect the yeast cells against the osmotic stress in the presence of 6% sugar. Next to ethanol and glycerol, a total amount of 0.16 ± 0.02% succinic acid and 0.05 ± 0.01% acetic acid were produced in the yeasted pastry samples, which was similar to the observation of Jayaram et al. [5] in bread dough (addition of 5.3% yeast). 

Since there is no sugar consumption by yeast in unyeasted pastry samples, no production of CO_2_, ethanol and acetic acid was observed in these samples, and only very low concentrations of succinic acid and glycerol were present at the end of the production process. These low acid and glycerol concentrations can possibly be attributed to microbial contamination [44,48].

### 4.2. Impact of Yeast Metabolites on Pastry Dough Properties during the Fermentation Phase

Yeast metabolites can affect pastry dough properties, such as dough height and rheology [4]. In unyeasted samples, no increase in height was observed during the resting step, whereas in yeasted samples, the height increased from 0.80 ± 0.00 cm to 2.65 ± 0.07 cm during fermentation. This increase in height resulted from the produced CO_2_ and possibly ethanol entrapped within the layered structure [29]. These results indicate that yeast has a significant impact on dough height during the fermentation phase of pastry making.

Changes in dough rheology were observed both in unyeasted and yeasted samples: during an incubation period of 75 min at 30 °C, the dough strength and elasticity decreased, while the dough extensibility increased. These dough rheology changes can be attributed to gluten relaxation, which is accelerated by the dough temperature rise during incubation (from 19 to 30 °C) [36]. Clarke et al. [49] attributed such rheology changes as a function of time to the breakdown of the large protein aggregates responsible for the dough’s structural integrity into small aggregates by the proteolytic activity of proteinases present in wheat flour or produced by bacteria contaminating the flour. 

In yeasted samples, this decrease in dough strength and extensibility was even larger compared to unyeasted samples as the result of the presence of large concentrations of various yeast metabolites [47]. The produced CO_2_, for example, not only increases the pastry height during fermentation, but also significantly reduces the dough density and hence the dough strength [50]. Ethanol, on the other hand, is known to increase the dough strength and elasticity and to decrease the dough extensibility [2,51,52]. Glycerol produced in the yeasted pastry samples probably also influenced the rheology of the dough. Aslankoohi et al. [47] observed increased dough extensibility, reduced dough strength and improved gas retention in bread dough when glycerol was added. Moreover, glycerol production by yeast can lead to changes in the metabolic fluxes resulting in changes in organic acid production [47], which can also impact dough rheology. 

In this study, organic acids reduced the dough pH from 5.75 ± 0.04 to 5.23 ± 0.10 during the production process. These results are in line with those obtained by Verheyen et al. [50], who found similar pH values in bread dough with 2.0% dry yeast, and by Jayaram et al. [5], who found that succinic acid is the major pH-determining yeast metabolite during straight dough fermentation. Because the pH optimum of the proteinases present in flour is around 4.4, it is possible that protein aggregates are broken down faster in the presence of organic acids [49,53]. Moreover, the presence of acids enhances the reduction of disulphide bonds, leading to reduced dough strength and elasticity and increased dough extensibility [49,54]. 

The lower dough strength and extensibility of yeasted dough compared to unyeasted dough in our study can thus be explained by the accumulated production of CO_2_, glycerol, ethanol and organic acids by yeast. These metabolites also have an influence on the optimal gas-holding capacity of fermenting dough [55], which is important for end-product quality. Next to the presence of yeast metabolites, the margarine layers in pastry dough could have an additional positive effect on the dough properties because they have their own strength and extensibility and can retain gas bubbles between the layers [37]. This retention of gas bubbles, for example, can lead to a lower dough density and hence a lower dough strength.

### 4.3. Fermentation Characteristics and Their Impact on End-Product Quality in Pastry Making

We showed in this study that sugar consumption and metabolite production affect the end-product properties of fermented pastry. In unyeasted samples, product height increased from 0.83 ± 0.10 cm to 1.78 ± 0.05 cm during baking due to evaporation of water and entrapment of steam within the layered structure [29]. These results are in line with those from Ooms et al. [56], who observed that dough lift was postponed until the dough temperature rose to 100 °C in unyeasted pastry samples. On the contrary, in yeasted samples, the increase in height during baking was significant but rather low compared to the height increase during fermentation. This increase during baking can be attributed to the evaporation of water and ethanol, decreased solubility of CO_2_ in the aqueous phase and expansion of CO_2_ in the gas bubbles [51,56,57,58]. Hence, these results confirm the hypothesis proposed by Ooms et al. [36] that the contribution of steam evaporation to dough lift is somewhat limited in fermented pastry production and that the fermentation phase is crucial for an acceptable end-product volume.

The remaining sugars in the end-product have a role in colour formation, due to Maillard and caramelisation reactions, as well as in sweetening. Yeast plays an important role in these reactions as it changes the absolute and relative abundance of the sugars present. According to the sugar analysis results, only a small part of the initially added amount of sucrose was consumed in yeasted pastry samples. The residual sucrose thus remains in the final product in the form of glucose and fructose, which are reducing sugars and can thus contribute to Maillard and caramelisation reactions. Indeed, a darker colour was observed in yeasted samples compared to unyeasted samples, which is an important determinant for consumer acceptance. Although the sugar concentrations are very different in yeasted and unyeasted pastry samples, the sweetness factor of these samples was not significantly different. This is because fructose has a higher relative sweetness (1.50) than sucrose (1.00). In unyeasted pastry samples, sucrose had by far the most significant contribution (96.0%) to the sweetness factor, whereas, in yeasted pastry samples, fructose had the most significant contribution (74.4%). 

### 4.4. The Impact of Different Sucrose Concentrations on Yeast Activity and Dough and End-Product Characteristics in Pastry Making

To evaluate the impact of different sucrose concentrations on yeast activity and dough and end-product characteristics in pastry making, fermented pastry samples were made with sucrose addition of 0, 7, 14 and 21%. Sugar and metabolite concentrations, CO_2_ production, dough rheology and dough and product height were measured. Samples with 14% (dm flour base) added sucrose are considered as reference samples because these samples are similar to commercially available fermented pastry products.

In samples without added sucrose, the CO_2_ production was high during the first 60 min of fermentation in the risograph, compared to samples with added sucrose. However, after 60 min, a slight decrease in CO_2_ production was observed because all glucose and fructose were consumed at this time point, forcing the yeast to switch to maltose consumption [1]. After 85 min, almost all sugars were depleted, causing a large decrease in CO_2_ production in the samples without added sucrose. When sucrose was added, no drop in CO_2_ production was visible because sugars were not limiting anymore. Increasing the sucrose addition gradually from 0 to 21% (on dm flour base) reduced yeast activity. This can be explained by the increased osmotic stress due to the higher sugar content [46]. The reduced yeast activity was visible in terms of a lower total sugar consumption amount and lowered CO_2_ and ethanol production. 

The total amount of CO_2_ produced during a fermentation time equal to the production time of fermented pastry samples was correlated with the sugar consumption during fermented pastry making (R^2^ = 0.97). The lower CO_2_ production in samples with 7% added sucrose than in samples without added sucrose can explain the higher dough strength and elasticity of these samples, because a lower CO_2_ production results in a lower dough density and, hence, a softer dough. Increasing the sucrose addition further to 14% did not have a visible impact on the dough strength and elasticity. Possibly, the impact of dough relaxation was too large to see the effect of the increasing sucrose addition [36]. In contrast to CO_2_ and ethanol production, glycerol production was higher in samples with added sucrose than in samples without added sucrose. This might indicate that glycerol production was increased in the samples with added sucrose to protect the yeast cells against the osmotic stress. 

Although less CO_2_ and ethanol was produced in samples with higher sucrose addition, the height of the samples with 0, 7 and 14% added sucrose was not significantly different after fermentation and after baking. We can assume that other factors than the gas-forming kinetics are also responsible for the product height [59]. It is, for example, known that the produced CO_2_ can destroy the integrity of the layered structures, resulting in the too early escape of steam during baking [28]. Increasing the sucrose level to 21% lowered the end-product height. This lower height might be caused by the higher amount of sucrose-water solvent [15]. When sucrose dissolves in water, it acts like a solvent, which impacts the gluten network. Changes in the gluten network can, in turn, affect the gas-holding capacity, resulting in changes in dough height [60]. These changes in the gluten network due to the high sugar content were in this study visible in the dough rheology measurements, as samples with 21% added sucrose had a lower dough strength and elasticity compared to the samples with 7 or 14% added sucrose. 

Next to product height, the colour and sweetness of the end-product were also impacted by a higher sugar concentration. With more sucrose, a darker colour, due to more Maillard and caramelisation reactions, and a higher sweetness factor were obtained. Another consequence of an increased sucrose addition was the lower release of maltose. Possibly, amylases are inhibited by high sugar concentrations, but more research is needed to understand these differences in maltose release. As described earlier, samples with 14% (dm flour base) added sucrose are considered reference samples representing commercially available fermented pastry products. High sugar content can, however, increase the risk of several health issues [21]. The results in this study can serve as a basis to explore sugar reduction strategies in fermented pastry products.

## 5. Conclusions

We investigated the dynamics of fermentation, the interplay between yeast and sugar and their impact on dough properties and end-product quality characteristics in fermented pastry making. The role of yeast and the fate of sugar during fermentation and baking has not been studied before in these multi-layered dough-fat systems. The role of fermentation in layered pastry products is partially different from that in other bakery products. The produced CO_2_ is, together with the layered structure, responsible for the characteristic texture of fermented pastry. The high sucrose concentrations are known to cause osmotic stress to yeast. We showed that yeast converts all sucrose into glucose and fructose in fermented pastry production, increasing the osmotic stress for the yeast and consuming part of the produced glucose. Together with the production of metabolites like CO_2_, ethanol, glycerol and organic acids, these sugar concentrations significantly impacted the dough and end-product properties. The produced CO_2_ increased the height, especially during the fermentation phase. The rheology of pastry dough was influenced by the cumulative effects of the presence of all these yeast metabolites. In future work, these results may be confirmed by adding these yeast metabolites to unyeasted dough. In yeasted samples, fructose had the most significant contribution to sweetness instead of sucrose. Next to the production of CO_2_, ethanol, glycerol and organic acids, yeast is also responsible for the production of many volatile compounds. In the future, it would be of interesting to analyse the production of these volatile compounds in fermented pastry products. Increasing the sucrose content led to higher osmotic stress, reducing sugar consumption and reducing CO_2_ and ethanol production. Moreover, increasing the sucrose addition resulted in a darker colour and a higher sweetness factor. Dough rheology was less affected by changes in sucrose addition, although no sucrose addition or a very high sucrose addition (21%) reduced the maximum dough strength. In the future, more research about the impact of the water-sugar solvent on unyeasted dough could, together with our results, provide more clarity about the distinct impact on dough rheology of the water-sugar solvent, on the one hand, and yeast activity, on the other hand. Based on the new insights from this study, further exploration of yeast-based strategies could help improve the production process or end-product quality. Reducing sugar content is of interest since high sugar intake levels are associated with an increased risk of many health problems.

## Figures and Tables

**Figure 1 foods-11-01388-f001:**
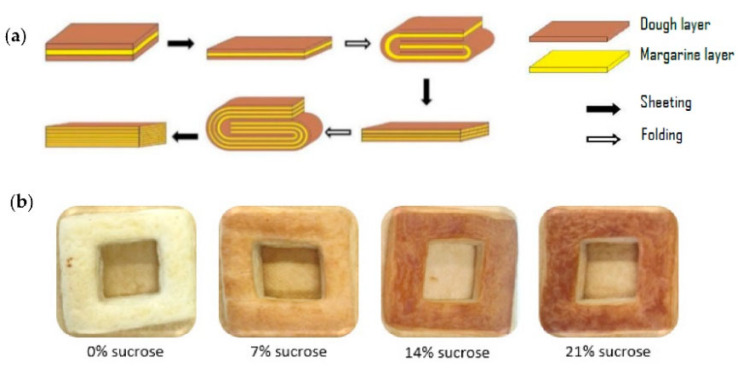
(**a**) Sheeting and folding process in the production of layered pastry dough [29]. Margarine layers (yellow) are folded into the predough (brown), resulting in one margarine layer and two dough layers. The dough is then repeatedly sheeted and folded until 27 fat layers are obtained. Reproduced with permission. (**b**) Baked end-product with 27 fat layers and 0, 7, 14 or 21% sucrose, made with a squared cutter.

**Figure 2 foods-11-01388-f002:**
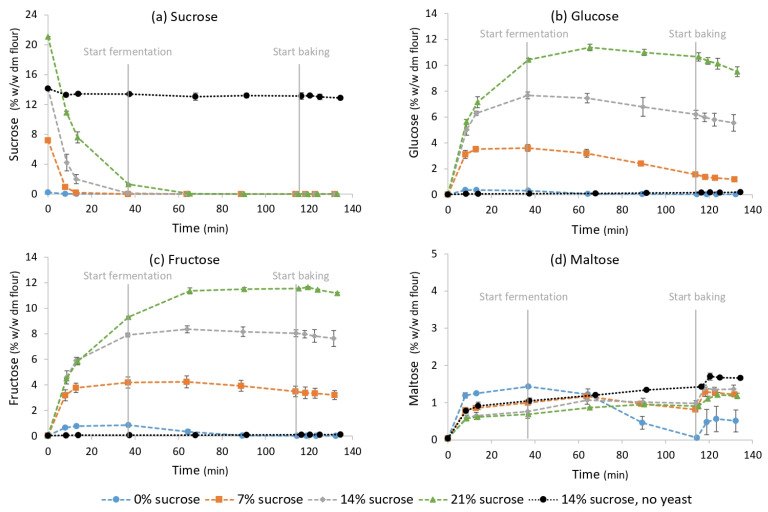
Evolution of (**a**) sucrose, (**b**) glucose, (**c**) fructose and (**d**) maltose concentrations in pastry samples with 8% yeast and 0, 7, 14 and 21% sucrose and samples with 0% yeast and 14% sucrose as a function of time (min) during pastry production. Concentrations are expressed on flour dry matter base (% *w*/*w* dm flour). The start of mixing is defined as t = 0. The end of the mixing phase, and the start of the fermentation and baking phases are displayed. Vertical bars represent standard deviations of duplicate measurements.

**Figure 3 foods-11-01388-f003:**
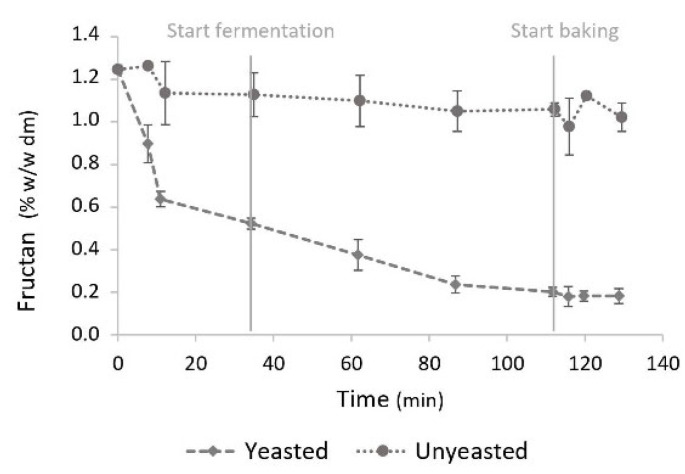
Evolution of fructan concentrations in pastry samples with or without yeast as a function of time (min) during pastry production. Concentrations are expressed on flour dry matter base (% *w*/*w* dm flour). The start of mixing is defined as t = 0. The end of the mixing phase, and the start of the fermentation and baking phases are displayed. Vertical bars represent standard deviations of duplicate measurements.

**Figure 4 foods-11-01388-f004:**
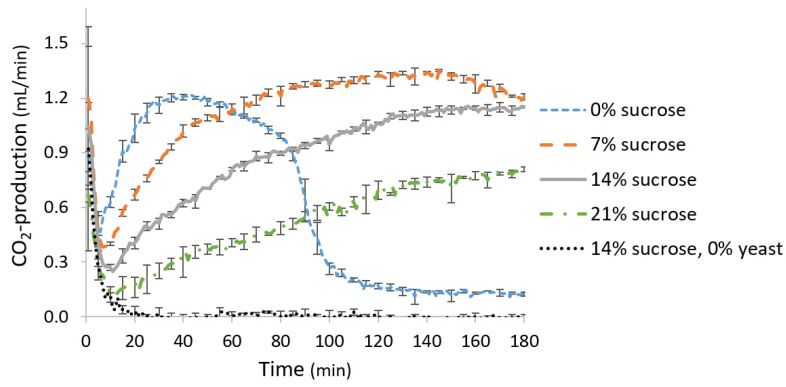
CO_2_ production rate (mL/min) in 10 g flour dough samples with 8% yeast and 0, 7, 14 and 21% sucrose and samples with 0% yeast and 14% sucrose as a function of time (min) at 30 °C. Vertical bars represent standard deviations of triplicate measurements.

**Figure 5 foods-11-01388-f005:**
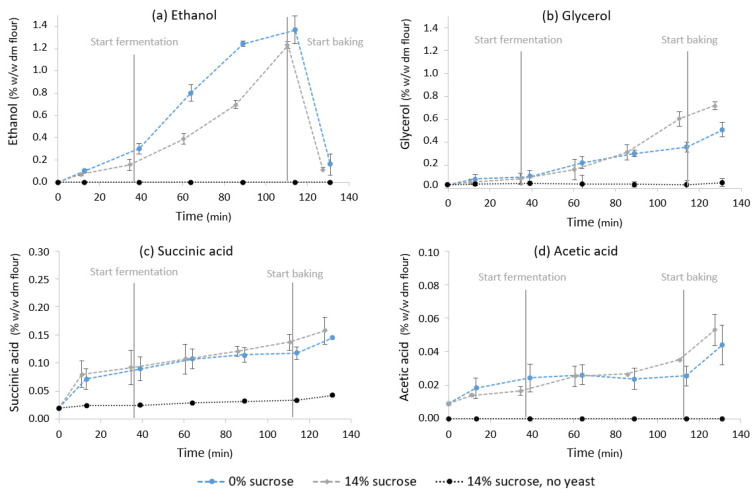
Evolution of (**a**) ethanol, (**b**) glycerol and (**c**) succinic and (**d**) acetic acid concentrations in pastry samples with 8% yeast and 0 and 14% sucrose and samples with 0% yeast and 14% sucrose as a function of time (min) during pastry production. Concentrations are expressed on flour dry matter base (% *w*/*w* dm flour). The start of mixing was defined as t = 0. The start of the fermentation and baking phases are displayed. Vertical bars represent standard deviations of duplicate measurements.

**Figure 6 foods-11-01388-f006:**
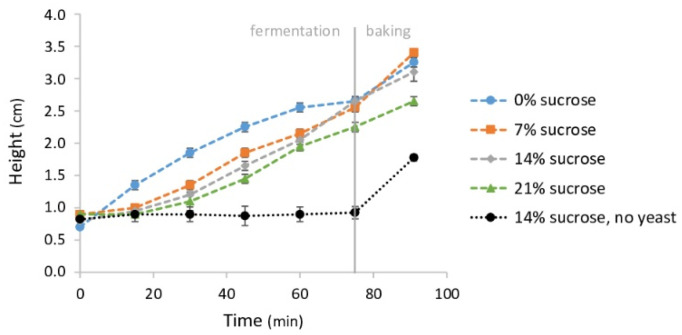
Height (cm) of pastry samples with 8% yeast and 0, 7, 14 and 21% sucrose and samples with 0% yeast and 14% sucrose as a function of time (min) during pastry production. The start of fermentation is defined as t = 0. Samples were fermented at 30 °C and 80% relative humidity for 75 min and baked at 220 °C for 16 min. Vertical bars represent standard deviations of duplicate measurements.

**Figure 7 foods-11-01388-f007:**
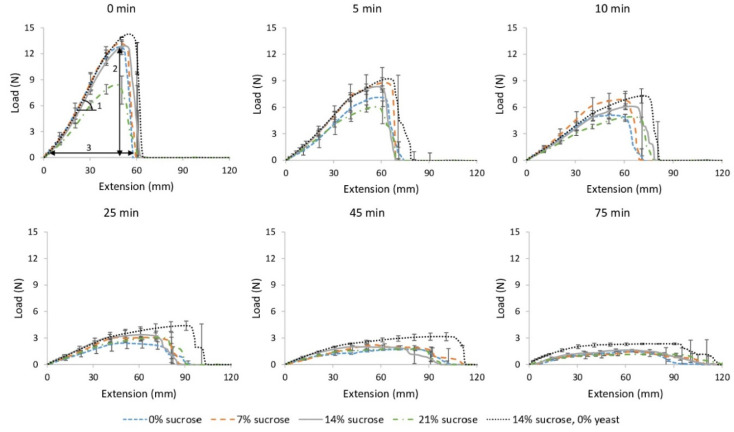
Evolution of dough rheology of laminated samples with 8% yeast and 0, 7, 14 and 21% sucrose and samples with 0% yeast and 14% sucrose during incubation at 30 °C and 80% relative humidity. Samples were taken before (0 min) and during (5–75 min) incubation. Load (N) is expressed in function of extension (mm). The slope of the linear part (1) represents dough elasticity (N/mm), the maximum height (2) represents dough strength (N) and the maximum width (3) represents dough extensibility (mm).

**Table 1 foods-11-01388-t001:** The individual calculated sweetness contributions (S_A_ = kC^n^) of glucose, fructose, sucrose, raffinose and maltose and the sweetness factor (S = S_A_ + S_B_ + …) for pastry samples with 8% yeast and 0, 7, 14 and 21% sucrose (% *w*/*w* dm flour base) and samples with 0% yeast and 14% sucrose. The standard deviations result from duplicate measurements.

	Glucose	Fructose	Sucrose	Raffinose	Maltose	Sweetness Factor
0% sucrose	0.00 ± 0.00 ^c^	0.00 ± 0.00 ^c^	0.00 ± 0.00 ^b^	0.00 ± 0.00 ^b^	0.11 ± 0.02 ^b^	0.11 ± 0.02 ^c^
7% sucrose	0.27 ± 0.01 ^c^	2.08 ± 0.15 ^c^	0.00 ± 0.00 ^b^	0.00 ± 0.00 ^b^	0.2 ± 0.00 ^ab^	2.55 ± 0.16 ^c^
14% sucrose	1.58 ± 0.24 ^b^	5.12 ± 0.54 ^b^	0.00 ± 0.00 ^b^	0.00 ± 0.00 ^b^	0.18 ± 0.02 ^ab^	6.99 ± 0.76 ^b^
21% sucrose	3.41 ± 0.63 ^a^	9.5 ± 1.61 ^a^	0.00 ± 0.00 ^b^	0.00 ± 0.00 ^b^	0.21 ± 0.04 ^ab^	13.12 ± 2.19 ^a^
14% sucrose, 0% yeast	0.02 ± 0.00 ^c^	0.02 ± 0.00 ^c^	7.15 ± 0.17 ^a^	0.01 ± 0.00 ^a^	0.25 ± 0.01 ^a^	7.45 ± 0.16 ^b^
*p*-value	<0.001	<0.001	<0.001	<0.001	0.022	<0.001

^a,b,c^ indicate a significant difference for the calculated sweetness of each sugar as analysed according to the global F-test (*p* < 0.05).

## Data Availability

No new data were created or analyzed in this study. Data sharing is not applicable to this article.

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
