# Peer review of "Sugar Levels Determine Fermentation Dynamics during Yeast Pastry Making and Its Impact on Dough and Product Characteristics"

_foods, 2022, doi:10.3390/foods11101388_

Round 1

Reviewer 1 Report

The manuscript presents complex and valuable information on the effects of yeast and sucrose on pastry dough properties. Generally, the manuscript is well written and presented. Some observations are made below.

L 241: ANOVA is used when there are more than 2 samples, while Student t test is used to compare 2 samples. I think there is a mistake in this row, please check.

Figures 4 and 7 are difficult to understand. I suggest to use colours. 

In the Discussion section there are many statements without citations. Also, for example 607-616 only presentation of the results is made. More comparations with the existing literature is recommended.

Where are the colour data? I did not find neither the method for determination of the colour.  

Reviewer 2 Report

The manuscript written by Timmermans et al. studies an interesting topic because there are not many works which investigate the role of yeasts during fermentation and their impact on sugars and on the rheological properties of dough.

The manuscript is well written by there are some observations to do.

Line 111: add the name of yeast, this is important because different strains have different properties.

Lines 168-171: add the manufacturer of the instrument and of the column used

Lines 386-387: the consumption of sugars is

In 14% sucrose: 1.79% glucose 0.49 fructose 0.14% maltose

In 7% sucrose: 2.64% glucose, 1.21% fructose, 0.3% maltose

In 21% sucrose 0.9% glucose

In total 5.3% glucose reduction, but in line 446 the authors indicate 23.6% of glucose consumed

This is unclear.

Moreover, due to this low consumption of sugar, it seems that the amount of CO2 and ethanol produced seems to be a little bit high

Finally, no repetitions of the experiments have been made: this is a mistake!

Reviewer 3 Report

The manuscript submitted by Courtin et al. mainly dealt with sugar levels determine fermentation dynamics during yeast pastry making and its impact on dough and product characteristics. The overview of this article is very well written and logistically good. And author tried to compare the sugar levels to find the fermentation difference in dynamics caused by various yeasts. Authors analyzed saccharide concentrations, CO2 production, dough strength and extensibility, sensory parameters such as sweetness, color, etc. However, an important odor description was missed. I think it's a pitiful design of this experiment. Besides this, some other problems or concerns should be paid much attention on the following aspects.

Abstract
You should add one sentence to indicate why did you make this experiment. I mean the background should put some drawbacks or shortcomings in this area, not just common statements.

Introduction
Some literatures about sensory changes related to dough fermentation should be added into the introduction. 

Materials and methods
line 113, Jiangsu should be Jiangsu, China
line 225, authors used calculation to get the sweetness of the end product, unfortunately, authors didn't use sensory evaluation to verify these results. I highly recommended the sensory panels' evaluation should be compensated here.

Results
line 256 to 257, these results might be compensated in your supplementary materials.
line 425 to 427, how did you get this result ? 

Discussions
line 453 to 457, In unyeasted pastry samples, only minor changes in sugar concentrations were observed. The slight decrease in sucrose and increase in glucose and fructose concentrations in unyeasted samples may be attributed to sucrose hydrolysis by microbial enzymes present in flour. Possibly, part of the released glucose and fructose is consumed by contaminating microorganisms or has participated in Maillard and caramelisation reactions during baking. what's the possible microbial enzymes and contaminating microorganisms ? authors need to clarify it.
line 489 to 532, authors need to compensate those volatiles changes during the fermentation.
line 563 to 606, the porisity and density of the dough should be discussed in this part.

Conclusions
limitations and further research plan should be added into this part.

Round 2

Reviewer 1 Report

The manuscript was improved according to the suggestions. 

L445: Since the colour determination was not performed instrumentally, I would suggest to move this section to the supplementary material or to rename it. Pictures used to discuss colour is not very "scientific".

Reviewer 3 Report

accept

Author Response

We want to thank the reviewer for assessing the manuscript.